# Distance to native climatic niche margins explains establishment success of alien mammals

Olivier Broennimann [1,2✉], Blaise Petitpierre[1], Mathieu Chevalier[1], Manuela González-Suárez [3], Jonathan M. Jeschke [4,5,6], Jonathan Rolland [7,8], Sarah M. Gray [9], Sven Bacher [9,10] & Antoine Guisan[1,2,10]

One key hypothesis explaining the fate of exotic species introductions posits that the establishment of a self-sustaining population in the invaded range can only succeed within conditions matching the native climatic niche. Yet, this hypothesis remains untested for individual release events. Using a dataset of 979 introductions of 173 mammal species worldwide, we show that climate-matching to the realized native climatic niche, measured by a new Niche Margin Index (NMI), is a stronger predictor of establishment success than most previously tested life-history attributes and historical factors. Contrary to traditional climatic suitability metrics derived from species distribution models, NMI is based on niche margins and provides a measure of how distant a site is inside or, importantly, outside the niche. Besides many applications in research in ecology and evolution, NMI as a measure of native climatic niche-matching in risk assessments could improve efforts to prevent invasions and avoid costly eradications.

[1] Department of Ecology & Evolution, University of Lausanne, Lausanne, Switzerland. [2] Institute of Earth Surface Dynamics, University of Lausanne, Lausanne, Switzerland. [3] Ecology and Evolutionary Biology, School of Biological Sciences, University of Reading, Reading, UK. [4] Institute of Biology, Freie Universität Berlin, Berlin, Germany. [5] Leibniz Institute of Freshwater Ecology and Inland Fisheries (IGB), Berlin, Germany. [6] Berlin-Brandenburg Institute of Advanced Biodiversity Research (BBIB), Berlin, Germany. [7] Laboratoire Evolution et Diversité Biologique, CNRS, Bâtiment 4R1, Toulouse, France. [8] Department of Computational Biology, University of Lausanne, Lausanne, Switzerland. [9] Department of Biology, Unit of Ecology & Evolution, University of Fribourg, Fribourg, Switzerland. [10]These authors contributed equally: Sven Bacher, Antoine Guisan. ✉email: olivier.broennimann@unil.ch

Investigating the factors responsible for the establishment success of alien species is crucial for understanding the proximate drivers of species invasions[1] and for designing management tools[2]. To establish in a new territory, a species must successfully pass through a series of filters:[3] it must be translocated there, find suitable abiotic conditions and resources to grow and reproduce, and withstand the new biotic settings in the invaded community. The abiotic environment is a critical filter, often considered through the prism of the species' environmental niche[4]. A pressing question in this regard is whether an introduction is more successful if it occurs at a site where the climatic conditions belong to the envelope of conditions experienced by the species within its native range (i.e., to the realized native climatic niche, NCN; including biotic interactions and dispersal limitations[4]). A traditional assumption is that the closer a site is to the NCN center, the greater the chances of successful establishment[5,6]. However, because realized niches are often asymmetrical[7], the NCN center does not necessarily describe a physiological optimum[8]. In such instances, the distance to NCN margins should be a better descriptor of how suitable a site is for establishment[7], because population fitness is expected to decrease towards the margins[4]. Surprisingly, although many studies have revealed that niche shifts can occur between the native and invaded ranges of alien species[9], none have investigated whether establishment must take place inside the NCN, or if it can also occur outside. So far, the importance of NCN-matching for establishment has remained largely untested, and no studies have used distance from NCN margins to explain establishment success.

We identify four main reasons for this shortage. First, due to the paucity of information on the success or failure of independent releases, NCN-matching is often measured at the scale of an invaded region[10,11] or at the scale of the whole invaded range, i.e., niche shift studies[9]. However, even if rare[12,13], niche shifts can emerge from (i) an initial successful establishment within the NCN followed by spread in novel conditions due to changes in biotic interactions or local adaptation[14], or (ii) a direct establishment outside the NCN (e.g., owing to competitive release) and further spread. Therefore, analyses comparing ranges (i.e., instead of introduction sites) to the NCN can yield misleading results concerning the importance of NCN-matching for establishment. Second, factors driving establishment success have mainly been examined at the species—rather than population—level[14], which implies a critical loss of information because the specificities of

independent release events are pooled. Tests of NCN-matching should rather be performed at the level of individual release events[15,16]. Third, national data are often used to quantify species' NCN, but the use of these geographically-restricted datasets may lead to niche truncation issues[17] that could cause NCN-matching underestimation[18]. For this reason, NCNs should be assessed globally to ensure taxonomically- and geographically- comprehensive NCN-matching. This has only been performed worldwide for reptiles and amphibians[11] and for birds[16]. Fourth, most NCN-matching metrics used to date have either used differences in latitude between the introduced and native range[19], inclusion in the same Köppen-Geiger climate class[20], climatic distance to the NCN center[16], or climatic suitability metrics derived from species distribution models[21]. However, while all these metrics inform about climatic matching inside the NCN, they are uninformative regarding how far a site lies outside the NCN (i.e., values are floored to zero).

Here, we aim to solve these issues by proposing the niche margin index (NMI) and by using it on a large release-event dataset for mammals including detailed information on the location and the number of released individuals[22]. NMI is the first niche metric that accounts for distance to niche margins (in and out). It is grounded in the classical theory that considers the niche as the response hypervolume in an environmental space where the population growth rate of a species at low abundance is positive if inside the niche envelope and negative if outside[6]. NMI thus provides a framework for the investigation of population fitness and source-sink dynamics in relation to niche margins[23], especially regarding population processes outside the niche[24,25], where evolutionary changes can potentially take place[6]. Concretely, NMI is a standardized ecological distance measured between a given site and the closest species' NCN margin (see methods). It can be calculated for all locations of interest (i.e., introduction sites in this study) and is standardized by the maximum distance to NCN margins in the study area to allow a comparison between species presenting different niche sizes (see methods). It thus ranges from $-\infty$ to $+1$, with negative values representing sites outside the niche (niche outerness), zeroes representing sites at niche margins, and positive values for sites inside the niche (niche innerness) (Fig. 1). NMI, therefore, allows a better assessment of populations outside the realized niche (or even outside the fundamental, i.e., within the 'tolerance niche'[25]), corresponding to non-self-sustaining sink populations[23] or observations in botanical gardens or parks[25] that should accordingly receive negative values of NMI. Extending the

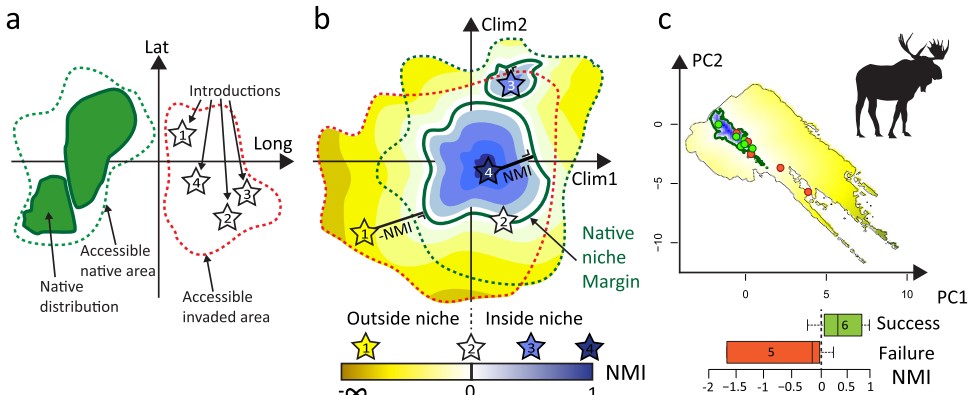

**Fig. 1 Illustration of the Niche margin index (NMI). a** Native distribution, accessible areas (sensu Barve et al. 2011) and alien introductions in geographical space. **b** Schematic representation of NMI with distances of introduction sites to native niche margins in the climatic space. **c** Illustration of NMI for the ungulate *Alces alces*. The native niche was estimated with a kernel density estimator (see "Methods"). Boxplots include the median (center line), the upper and lower quartiles (box limits), the 1.5× interquartile range (whiskers). $N = 11$ independent introductions. Drawings and schematic representations are made by the authors.

habitat suitability gradient toward negative values might thus also help with finding a more significant relationship between habitat suitability and population fitness[26]. Finally, as reduced population fitness outside the realized niche could also largely be caused by biotic interactions, the NMI additionally offers a new way of integrating niche measures, biotic interactions, and coexistence theory in a common framework[24].

We use NMI to estimate the climatic-matching of 979 introduction sites to the NCN of 173 alien mammal species. We assess to what degree NMI can explain establishment success, alone and in complement to life-history attributes and historical factors commonly considered in this context[10,15,27]. For this purpose, we use a Bayesian hierarchical framework from which we extract posterior distributions of model parameters to assess the strength (median of the posterior distribution) and confidence (measured as the 95% Highest Posterior Density interval [HPD$_{95}$]) of the effect of life-history attributes (litter weaning age, litter size, number of litters per year, coefficient of variation of adult body mass and coefficient of variation of neonate body mass), historical factors (native range area, introduction date, number of introduced individuals, and introduction on mainland vs. island), and NMI on establishment success. To account for non-independence in the data, we include random effects for the taxonomy and the biogeographic region of introduction (see Methods). We show that NMI alone significantly explains establishment success, and that its effect in Bayesian models is stronger and more consistent than that of most other previously studied life-history and historical factors.

## Results and discussion

**NMI explains establishment success.** Establishment was more successful when the introduction sites were inside the NCN, i.e., with higher NMI values. This was the case when analyzing NMI alone (Wilcoxon rank-sum test: $U = 59822$, $p < 1 \times 10^{-6}$, Fig. 2) or in combination with life-history attributes and historical factors (Bayesian approach: posterior $P[\text{NMI} > 0] = 99.7\%$, effect

size $= 0.31$, HPD$_{95} = [0.01; 0.55]$, Fig. 3; see also phylogenetic regressions: Supplementary Note 2, Supplementary Fig. 3). Overall, 69.6% (682 out of 979) of introduction sites had positive NMI values, indicating that most individuals were introduced within their NCN. This was true for both successful (557 out of 787) and failed (105 out of 192) establishments. When tested individually, most species showed higher NMI values for sites where establishment was successful, but differences were not always significant, likely due to limited statistical power (Supplementary Fig. 2).

**NMI vs. life-history and historical factors.** Investigations of alien establishment success in mammals have focused primarily on life-history attributes (e.g., brain size[28], reproductive lifespan[27], phenotypic plasticity[15]), historical factors (e.g., founding population size[29], time since introduction[15]), or characteristics of the recipient community (e.g., number of alien species already introduced[30]). Results for many of these factors are either inconclusive or highly dependent on the taxonomic group and the study area[30]. The number of introduced individuals appears to be the most consistent predictor of establishment success across studies[31], both in terms of number of introduction attempts and the number of individuals per attempt[6]. Here, Bayesian inferences showed that NMI had a higher posterior probability for a positive effect ($P[\text{effect} > 0] = 99.7\%$), and a larger effect size (0.31) on establishment success than most of the historical factors and life-history attributes classically considered in this context (Fig. 3). Only the number of introduced individuals showed stronger evidence for a larger effect size than NMI (Fig. 3; $P[\text{effect} > 0] = 99.7\%$; effect size $= 0.56$; HPD$_{95} = [0.15; 1.00]$). The number of litters per year and the coefficient of variation of neonate body mass also presented positive effects on establishment success ($P[\text{effect} > 0] = 83\%$ and 95%; effect size $= 0.27$ and 0.36, respectively), but confidence was lower (HPD$_{95} = [-0.27; 0.82]$ and $[-0.07; 0.82]$, respectively; Fig. 3). These traits may however have a direct impact on population growth by

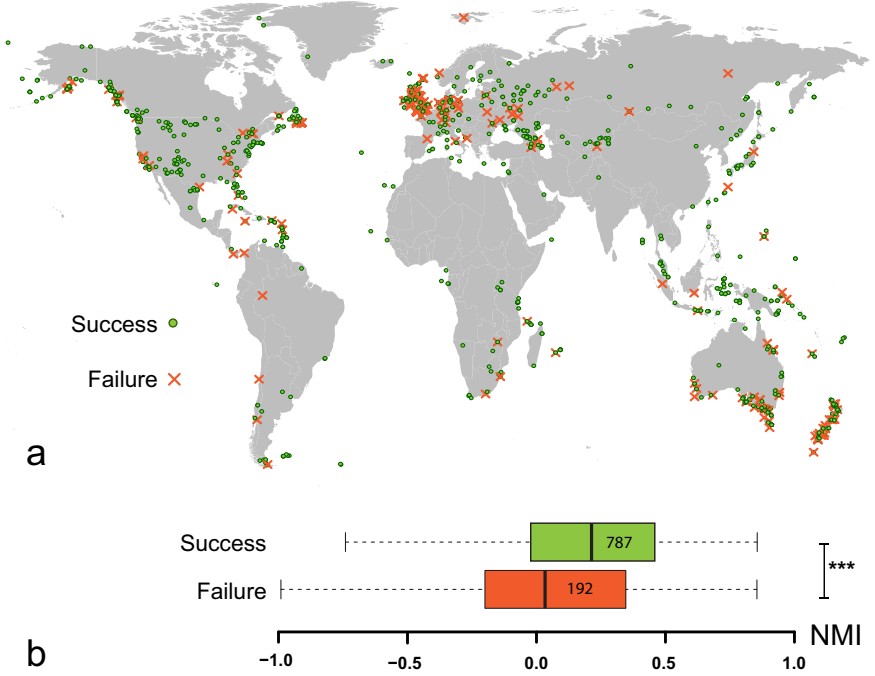

**Fig. 2 NMI and establishment outcome. a** spatial distribution of successful and failed introductions. **b** NMI is significantly higher in successful establishments (Wilcoxon rank sum wo-tailed test: $U = 59822$, $p < 1 \times 10^{-6}$). Boxplots include the median (center line), the upper and lower quartiles (box limits), the 1.5× interquartile range (whiskers). $N = 979$ independent introductions.

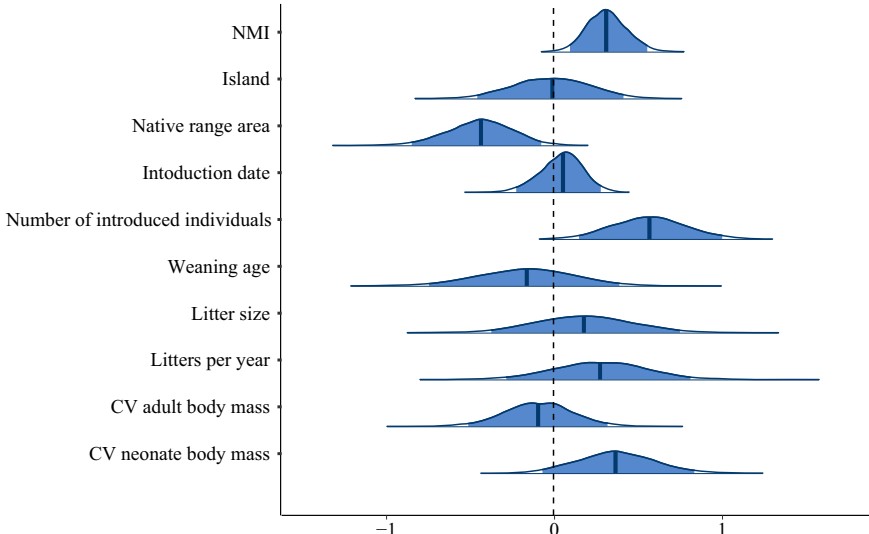

**Fig. 3 Posterior coefficient estimates of the Bayesian model.** Distribution of coefficients for fixed effects is shown. All variables in the analyses are standardized to mean zero and unit variance implying that coefficients indicate effect sizes that can be directly compared among variables. Vertical blue lines represent the median of the posterior distribution of effects (i.e., the strength of effect), while the blue shaded areas under the curves represent the 95% Highest Posterior Density (HPD) intervals. The vertical dotted line indicates no effect. $N = 3000$ independent samples from the posterior distribution of model estimates. The posterior probability for a positive effect of the NMI on establishment success (P[NMI > 0]) equals 99.7%.

reducing extinction probability for small populations[29]. The native range area had a negative effect on establishment success (Fig. 3; P[effect>0] = 99.1%; effect size = −0.43; $HPD_{95}$ = [−0.85; −0.08]), possibly because species with a narrow native range can be heavily constrained by geographic barriers, leading to an underestimation of their NCN. Beyond confirming the previously shown importance of reproductive investment and number of introduced individuals, we show that NCN-matching ranks among the most important variables to explain establishment success. The importance of climatic niche matching was also supported using a traditional model-based climate suitability index (i.e., predictions obtained from species distribution models; see Supplementary Note 1; Supplementary Data 1; Supplementary Figs. 4 and 6; see also[16]) but at a higher computational cost and the loss of the niche outerness information. Our findings thus support a systematic use of NMI-based climate matching (or equivalent niche metrics able to inform how far a site lies outside the NCN) in studies assessing establishment success.

**Mammal introductions occur mostly within the NCN.** NMI provides key insights for sites located inside (niche innerness) but also outside (niche outerness) niche margins; a characteristic never assessed before. Our results show that most releases of mammals occurred inside the NCN, even for failed establishments. The limited number of failed establishment outside the NCN in our dataset is likely because most reported introductions of mammals were deliberate (88.1%) and people releasing animals probably had an intuitive understanding that climate plays a role in establishment success, and thus likely avoided introducing species in climatically unsuitable areas. Moreover, it is difficult to document failed establishments in retrospective analyses of accidental introductions, as generally we become aware of only those invasions that were successful[6,32]. If all introductions had been accidental, a larger proportion of sites may have fallen outside the NCN and the signal we detect would have been even stronger.

**Successful establishments outside the NCN.** Niche theory predicts that a population can only establish if introduced within the NCN, i.e., within positive NMI values[6]. Even if we show here that this is generally the case, some introductions (24%; Supplementary Fig. 11) were nevertheless successful outside the NCN. This could be due to methodological problems. We may for instance have missed climatic (e.g., extreme climatic conditions) or ecological (e.g., habitat use) predictors that may influence the size and the shape of the NCN for some species. The internal predictive capacity of the Bayesian model (maxTSS of 0.52) on a 20% left-out evaluation dataset suggests that the model could be improved by the use of missing predictors. These predictors are likely to be species-specific, calling for more detailed investigations at the species level. However, and most importantly, evidence of successful establishment in niche outerness situations could also reveal areas where particular evolutionary or ecological processes occur within local populations[6]. For instance, it is possible that biotic interactions, limited available conditions and dispersal limitations in the native range imply that the NCN only provides a partial overview of the climatic conditions that species can tolerate[33]. Therefore, some populations might have successfully established outside the NCN due to a release from competition[24,34], to preadaptation to conditions not present in the native range[35], or to niche evolution[6,32]. While outside the species' realized NCN, are most of these populations still inside the fundamental niche? To investigate this question, we ran a complementary analysis using minimum volume ellipsoids to delineate the NCN (MVE; see methods, Supplementary Fig. 11). MVEs, which are centered on the mass center of observed occurrences and which long and short axes are calibrated to include a given level of observed occurrences, have been proposed as an estimate of the fundamental niche[36]. We found that 19% of successful establishments were located outside the realized niche but inside the putative fundamental niche. Only five percent were located outside of both the realized and fundamental niche (i.e., part of the "tolerance niche"[25]), potentially indicating cases of populations being self-sustaining only temporarily or due to

human facilitation (e.g., in zoological parks). These results support the classic view that most successful establiments outside the NCN are due to a release from competition[24,34]. Note that in line with the "tolerance niche" concept, if a given site presents a negative NMI in current climatic conditions but a positive (or less negative) value in the future, this would indicate that establishment success at this site is more likely in the future. NMI could thus be used as a powerful indicator of where climate change could increase invasion risks[37].

**NCN-matching as a tool to anticipate risk of biological invasions.** Introduction followed by initial establishment is a critical step in the invasion process: once alien species are established, control or eradication becomes difficult and costly[38,39]. Prevention is thus the most cost-effective measure against alien species[39]. Here, we showed that most alien mammals worldwide are more successful at establishing when the climate matches the conditions of their native niche, thus confirming the corresponding, but rarely tested hypothesis[16,33]. A recent study on invasive birds showed that a simple measure of NCN-matching (based on the mean and standard deviation of climate conditions across grid cells in the native range) significantly explains establishment success, but with relatively low effect size compared to life-history attributes and historical factors. Here, using the NMI metric, we show that, for mammals, NCN-matching stands among the main predictors of establishment success. The establishment success of exotic animals thus appears to depend on the particular combination of species attributes and release event characteristics[34,40], which explains why general features of invasions have been difficult to characterize[41]. We show that it is nonetheless possible to predict establishment success using NCN-matching, together with the number of introduced individuals and possibly life-history attributes related to reproductive investment. NCN-matching should therefore be systematically included in pre-border invasion risk assessments of environmentally- and socio-economically-damaging biological invaders[42].

## Methods

**Introduction data.** From the sections "History of Introductions" for introduced mammals presented in[22], we extracted the (1) geographic coordinates, (2) number and date of release of introduced individuals and (3) the outcome of each introduction event. John Long compiled this information over a period of 31 years (between 1969 and 2000) from an impressive list of references including peer-reviewed publications, books, governmental and nongovernmental reports and conference proceedings all around the world and in many languages. To our knowledge, this is the most complete global dataset on mammal introduction outcomes available. Only populations that were explicitly described as established or expanding were considered as successfully established. Reintroductions in historical parts of the native range of species were not considered. Only species that had no major documented contractions of their range extent during historical times were considered for analysis (but range fragmentations were allowed). This assessment was based on the description of the species in the IUCN Red List database (iucnredlist.org; accessed in November 2013), in particular on information from the "range description" section. We recorded all introduction events for species with fewer than 20 known introductions and the 20 first for species with more than 20 introductions. We chose this threshold of 20 introductions per species to keep the digitalization work manageable, while keeping all species and a reasonable amount of variability in introductions within each species. It resulted in a database of 979 introduction events for 173 mammal species. The number of introduction events varied from one to 36 depending on the species (mean = 5.58; SD = 7.16). For every introduction, we gathered information about historical factors at the release-event level, and life-history attributes at the species level commonly considered in this context[10,15,27]. Historical factors included introduction date (digitized from[22]), number of introduced individuals (digitized from[22]), introduction on mainland vs. island (extracted from GADM data shapefile; gadm.org), native range area (extracted from IUCN Red List database shapefile; iucn-redlist.org) and the biogeographic region of introduction (extracted from the shapefile of terrestrial ecoregions; maps.tnc.org/gis_data.html). Life-history traits included litter weaning age, litter size, number of litters per year, coefficient of variation of adult body mass, and coefficient of variation of neonate body mass[15].

**Native climatic niches (NCN).** The native ranges of mammal species for which we had introduction events were extracted from the IUCN Red List database (iucn-redlist.org) in November 2013. Using the information contained in the attribute table of the shapefiles, we considered areas labeled as "extant", "probably extant", "reintroduced", "probably extinct" and "extinct" as part of the native range. We used expert-based IUCN range maps to quantify native niches instead of occurrences from GBIF (Global Biodiversity Information Facility; www.gbif.org) because they include areas where the species has disappeared but might be important to quantify the whole native climatic niche[43]. The use of GBIF data to quantify niches is further challenged by the fact that these data are spatially biased and prone to identification errors. To quantify the native niche (NCN) of the species, we used eight bioclimatic variables available worldwide at a resolution of 10': daily temperature range, temperature seasonality, temperature of the coldest quarter, temperature of the warmest quarter, precipitation of the driest quarter, precipitation of the coldest quarter, and precipitation of the warmest quarter from worldclim variables (bio2, 4, 10–11 and 16–18 from[44]) and annual aridity (ai, from[45]). These variables are commonly used to quantify the niche of invasive species at global scales (e.g., [46]). We calibrated a principal component analysis (PCA) using the values of all pixels worldwide for the climatic variables (i.e., as the "PCAenv"[47]). This PCA provides a reduced climatic space that maximizes the climatic variation along principal component axes (Fig. 1B in the main text). The two first axes of the PCA explained 78.33% of the variation and were subsequently used to define the climatic space for the niche quantifications of all species (Supplementary Fig. 1). Note that more axes could be retained to increase the percentage of variation explained in the PCA, but given the high explanatory power of the first two axes, we decided to keep only these two for the ease of representation and interpretation. For each species, we calculated the PCA scores of the pixels belonging to the native niche according to the IUCN range maps. We then used a kernel density estimator (function *kde*; R package *ks*[48]) to extract the contour line corresponding to 99% of the estimated density of occurrence along the PCA axes, thus creating a representation of the native niche in the climatic space (NCN; see Fig. 1B for a schematic representation, Fig. 1C for an example with *Alces alces*). To check the sensitivity of our results to the level of estimated density of occurrence, we repeated our analyses with contour lines corresponding to 95 and 90% (Supplementary Figs. 7–10). Furthermore, because IUCN range maps describe the broad outlines of a species distribution[49], the species may not occur at any given location within it. Rasterizing these maps at too fine of a resolution may wrongly associate particular sets of climatic conditions—typically mountain tops - to a species' niche. We, therefore, repeated our analyses with two coarser resolutions (0.5° and 1°; Supplementary Figs. 7–10). Finally, when a species is introduced to a new continent, it is hypothesized that populations can grow and reproduce in a part of the fundamental niche larger than indicated by the realized niche of the species, e.g., due to release from competition. Some authors have proposed using minimum volume ellipsoids (mve) to obtain closer estimates of the fundamental niche, by allowing volumes that can encompass parts of the environment unavailable now but that potentially existed in the past[36]. We therefore also ran all analyses using mve instead of kde envelopes (Supplementary Figs. 8–10). Similar results were obtained under these different settings, strengthening our conclusions and the utility of the *NMI* to explain establishment success.

**Niche margin index (NMI) as a measure of NCN-matching.** We developed a metric that we called niche margin index (NMI; Fig. 1) to measure how far the climatic conditions present at the introduction sites are inside or outside the NCN. The calculation of NMI implied three steps. (1) We assigned a positive sign to NMI if the introduction sites were located inside the NCN, and a negative sign if outside the NCN. (2) We calculated the minimum orthogonal distance of introduction sites to the NCN margin using the gDistance function of the R package rgeos. Note that this distance is measured here on a plane, but gDistance can measure distances in more than two dimensions if more PCA axes are included in the analysis. For single-species studies, this distance could be used directly. However, in multi-species studies, NMI should be standardized to allow comparison between species presenting different niche sizes. (3) We scaled each distance by the maximal orthogonal distance to the margin from anywhere inside the NCN (i.e., the distance to the margin from the "centroid", defined here as the most distant point from the margin inside the NCN). To achieve this, we generated 10,000 points regularly spaced inside of the NCN using the function spsample (R package sp) and for each of these points, we calculated the minimum orthogonal distance to the margin. Among all these minimum distances, we selected the longest one and used it as a denominator to scale the distance calculated at step 2. This standardization by the "maximal minimum orthogonal distances" ensures that populations located inside the NCN cannot take a NMI value higher than 1 and that values are comparable across species (i.e., a value of 1 indicates the location the further away from NCN margins, regardless of the size of the NCN). Note however that outside of the NCN, values can be smaller than −1 (when the distance for a site is larger than the distance used for standardization). NMI is thus a standardized ecological distance ranging from −∞ to +1 that measures the distance of an introduced population at a given site (or of any site) to the NCN margin of a species in a climatic space, here defined by the first two axes of a PCA (Fig. 1B). A NMI value of +1 indicates that the population was introduced at a site with climatic conditions corresponding to the center of the NCN, a NMI value of 0 characterizes locations with climatic

conditions at NCN margins, while a highly negative value reflects an introduction at a site where climatic conditions are far outside the NCN.

**Statistical analyses**. The relationship between establishment success and NMI was first assessed alone using a Wilcoxon rank-sum test (also known as Mann–Whitney test) and then in conjunction with other explanatory variables commonly used in studies of introduction success (see[15,27] and references provided in the Introduction section of the main text) using a hierarchical Bayesian mixed-effect model.

*Model structure*. We investigated the effects of NMI, species attributes (native range area, weaning age, litter size, litters per year, the coefficient of variation of body mass and the coefficient of variation of neonate body mass), and historical factors (introduction date, number of introduced individuals, and a binary variable indicating whether introductions took place on an island or on the mainland) altogether using Bayesian inference. Specifically, establishment success at site $i$ for species $s$ (denoted $Y_{i(s)}$ to reflect the fact that sites are nested within species) was assumed to follow a Bernoulli distribution with success probability $\psi_{i(s)}$:

$$Y_{i(s)} \sim \text{Bern}(\psi_{i(s)}) \tag{1}$$

On the logit scale, $\psi_{i(s)}$ was modeled as:

$$\text{logit}(\psi_{i(s)}) = \alpha + \alpha_{s(fam)} + \alpha_{fam} + \alpha_{region} + \beta \times \text{NMI}_{i(s)} + \sum_c^9 \beta c \times \text{Covariate}_{c,i(s)} \tag{2}$$

where $\alpha$ is the main intercept (i.e., average success probability on the mainland) while $\alpha_{s(fam)}$ are species-wise intercepts nested within family-wise intercepts (denoted $\alpha_{fam}$) and $\alpha_{region}$ are intercepts associated to biogeographic regions. These three random effects were assumed to be normally distributed with means of zero and standard deviations $\sigma_s$, $\sigma_{fam}$, and $\sigma_{region}$, respectively. The parameter $\beta$ is a slope coefficient representing the effect of the niche margin index (NMI) while parameters $\beta_c$ are slope coefficients representing the effect of each of the nine above-mentioned covariates ($\text{Covariate}_{c,i(s)}$) on establishment success. $\text{Covariate}_{c,i(s)}$ is a two-dimensional array containing the value of species attributes and historical factors $c$ measured for introduction event $i$ and species $s$. Note however that for some covariates (mostly species life-history attributes) we did not have information about variation across sites and thus these covariates are assumed to be fixed at the species level. The number of introduced individuals, weaning age, native range area, introduction date and litter size were log-transformed before analysis to reduce the skewness of their distributions[31]. All continuous covariates were standardized to z-scores (mean of zero and standard deviation of one) before analysis. The correlation between covariates varied from −0.8 to 0.48.

To check the robustness of our results we ran an additional model where we replaced NMI by a measure of climatic suitability (CS) obtained from species distribution models (Supplementary Note 1). Very similar results were obtained with this model (Supplementary Fig. 6).

*Parameter estimation*. Posterior samples of model parameters were obtained by MCMC sampling with JAGS[50] run through the R environment[51] using the package R2jags[52]. The model was run with two chains with a burn-in of 5000 and an additional 20,000 iterations with a thinning interval of 20 iterations. For each chain, initial values were randomly selected in different regions of the parameter space.

For fixed intercepts and slope coefficients, we used normal prior distributions with means zero and precision of 0.1 (equivalent to a standard deviation of about 3.1). For standard deviations associated to random intercepts ($\sigma_s$, $\sigma_{fam}$, and $\sigma_{region}$), we used half-Cauchy distributions[53]. Because some covariates had missing values for some species or sites, we generated new data using Bayesian imputation. Doing so allowed us to conserve and use the information for the other covariates, while producing neutral estimates for the missing values. We followed Missing Completely At Random procedure, thus assuming the location of missing values in the covariate matrix is completely random with respect to other values[54]. In practice, covariates were assumed to follow a normal distribution with mean $v_c$ and standard deviation $\sigma_c$ (with c varying from one to nine). The priors used for $v_c$ were normal distributions centered on zero with standard deviations of 10, while half-Cauchy distributions were used for $\sigma_c$.

For each parameter, effect sizes were estimated as the median of the corresponding posterior distribution while confidences were quantified using the 95% HPD interval (function *boa.hpd*; R package boa). For all parameters, we evaluated the posterior probability for the observed effect size (median of the posterior distribution) by calculating the proportion of samples from the posterior distribution displaying the same sign as the observed effect size. For instance, for a positive effect size, we calculated the posterior probability for a positive effect (P [effect>0]) by dividing the number of MCMC samples with a positive sign by the total number of MCMC samples. Accordingly, a value of e.g., 0.8 would indicate that the posterior probability for a positive effect is 80%.

*Convergence, fit, explanatory and internal predictive power of the models*. Convergence was assessed for all parameters using the Gelman and Rubin convergence diagnostic with a threshold fixed to 1.1[55].

As posterior predictive checks, we use the sum of squared standardized Pearson residuals[56]. This metric was calculated for both the observed data and a replicated dataset derived from model estimates. From the obtained values, we quantified the proportion of MCMC samples in which the distance of observed data to the model is greater than the distance of replicated data to the model (i.e., the so-called Bayesian P value). Values close to 0.5 suggest a good model fit, whereas values close to 0 or 1 indicate a lack of fit.

The explanatory power of the models was evaluated on the full dataset using the true skill statistic (TSS[57]) across all possible thresholds between 0 and 1 (maxTSS[58]). The internal predictive performance was evaluated using a repeated split-sample approach, with 80% of occurrence records used for training the model and 20% used for evaluation. Models were evaluated using the maxTSS. This procedure was repeated 10 times.

The model presented no evidence of convergence problems (potential scale reduction factor below 1.1 for all parameters) and posterior predictive checks revealed no indication for a lack of fit (Bayesian $p$ value = 0.22; See Supplementary Fig. 5 for detailed results). The explanatory power of the model was good overall (maxTSS = 0.52; Supplementary Fig. 10). The predictive power of the model was also good with maxTSS ranging from 0.31 to 0.5 (Supplementary Fig. 10).

**Reporting summary**. Further information on research design is available in the Nature Research Reporting Summary linked to this article.

## Data availability

The introduction dataset, including information about establishment success, life-history traits, historical factors, and release characteristics, is available on github.com/ecospat/NMI[59]. Raw data on establishment success are shown in Fig. 2a. Source data are provided with this paper.

## Code availability

The code to generate NMI values for introductions from IUCN native range maps and to perform the Bayesian mixed model is also provided on github.com/ecospat/NMI[59]. The code also allows readers to replicate Fig. 1c, Fig. 2b, and Fig. 3.

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

## Acknowledgements
The original idea of the manuscript was developed in the project "Plant Survival" funded by the National Centre of Competence in Research (NCCR), a research program of the Swiss National Science Foundation. J.R. acknowledges funding from the LABEX TULIP and CEBA.

## Author contributions
O.B.: co-designed the study, gathered and digitized the introduction data, conceived the innerness index with B.P. and A.G., analyzed the innerness, ran the mixed models, and wrote the initial draft of the paper with B.P. B.P.: co-designed the study, gathered the environmental and distributional data, conceived the innerness index with O.B. and A.G., measured innerness and wrote the initial draft of the paper with OB. M.C.: performed the Bayesian analyses and contributed to write and revise the paper. M.G.: provided species' trait data and contributed to write and revise the paper. J.M.J.: provided species' trait data and contributed to write and revise the paper. J.R.: performed the pgls analyses and contributed to write and revise the paper. SMG: contributed to write and revise the paper. S.B.: co-designed the study, supervised the analyses and contributed to write and revise the paper. A.G.: co-designed the study, conceived the innerness index with B.P. and O.B., supervised the analyses and contributed to write and revise the paper.

## Competing interests
The authors declare no competing interests.
