## [Peer Review File · Nature Communications]

Reviewers' Comments:

Reviewer #1:

Remarks to the Author:

Species distribution models have become a common tool to predict sites at risk of invasion by introduced species. Yet, there is an ongoing debate about the reliability of these models, as studies frequently report mismatches between actual and predicted invasive occurrences. These mismatches are often interpreted as so-called 'niche shifts', calling into question the generality of the hypothesis of niche conservatism ('NCN', as termed by the authors) that underlies the use of species distribution models for forecasting invasive ranges.

The authors here point out that part of that part of the debate may stem from the fact that the vast majority of published studies (necessarily) relies in invasive range occurrence data that represent a mix between areas that are occupied by invasive species because (a) species have been introduced there and they have successfully established a self-sustaining population, and (b) invasive spread from another location is the source of the invading individuals, not a new, independent release event. For case (b), it can be argued that invasive spread may result from e.g. adaptation to novel local climates, and that including such data when evaluating conservatism of the native niche is not a 'fair' comparison. I think this is a very valuable remark, and commend the authors for gathering the data necessary to make a 'fair' assessment of the NCN.

The authors then proceed to with a fair test of the NCN hypothesis using both a novel Niche Margin Index (NMI, see below) and more traditional Climate Suitability (CS) index derived from a species distribution model. Results were similar, evidencing that introductions are more likely to succeed when the area in which they were introduced had a climate that was more similar to the climate conditions species experience in their native range. This result corresponds with those of a recent meta-analysis of NCN studies (i.e. those studies that mix 'establishment' and 'invasive spread' occurrences'), which concludes that by and far, invasive species do conserve their climatic niche (Liu et al. 2020, PNAS, <https://doi.org/10.1073/pnas.2004289117>). For anticipating biological invasions, it is good news that both establishment success *sensu strictu* (and invasive spread seemingly too, Liu et al. 2020) are tightly linked to the NCN. One caveat may here however be that the authors use a rather liberal definition of a species native niche, reducing the chance that a non-native occurrence will fall outside of a species niche (see below, technical comments).

Assessing NCN has seen some methodologies put forward over the last decade, varying from metrics based on species distribution models or ordination frameworks, aimed at testing hypotheses on niche equivalency and similarity or on the proportion of species occurrences that are within or outside of a species native niche. While in theory these methods can be applied to 'establishment data' only, the authors here introduce an interesting novel metric, the NMI, that allows for each pixel in a geographical area to be assessed in terms of its 'distance' to the margins of the species native niche (negative outside, 0 at the margins, positive inside the niche).

While certainly an interesting metric, especially the 'niche outerness' component, the discussion of how this niche outerness can advance the field of biological invasions, both from a theoretical and applied perspective is a bit scant, in my opinion. The authors do mention that 'particular evolutionary or ecological processes [can] occur', but as an indication of climatic suitability under climate change, but little effort is made to relate their metric to concepts such as the 'tolerance niche', as defined by Sax et al. (2013, *TREE* 28: 517-523), or with potential relationships between niche distances and population growth rates given in Holt et al Chapter 10 *Theories of Niche Conservatism and Evolution: could exotic species be potential tests?* pp 259-290 in *Species Invasions: Insights into ecology, evolution and biogeography* (eds. Sax, Stachowicz and Gaines), Sinauer Associates, Mass. This is a bit of a missed chance to showcase the relevance of this novel metric.

Apart from these general remarks, I have only very few more technical comments, as the paper is

overall well written, clearly structured and easy to follow.

Abstract: maybe a bit nitpicking but I find the formulation that to be a successful invader, a species needs to establish in conditions within its native climate niche a 'bit upside down'. To be successful, all an invasive species needs to do is to establish a (self-sustaining) population. The hypothesis goes that such successful establishment will only be possible in areas that are within its native climatic niche.

Line 203-213: Were all occurrences (here: all pixels within native range) used to delineate the niche margins, as I see two potential problems here. (1) of the niche margins encompass even occurrences in very rare climate conditions, this may result in rather broadly, liberally defined 'native niches', making it more likely that invasive occurrence will fall within the native niche. Have you considered sensitivity analyses, e.g. calculating niche margins when omitting the, say, 5% occurrences that are in the most uncommon climates (climates here as 'pixels' in the 100-by-100 climate grid)? (2) This seems especially relevant for this study, as not Shapefile range maps have been used for characterizing the native distribution and thus the native niche. Range maps often do a pretty good job at describing the broad outlines of a species distribution, but this does not mean that species will occur at any given location within that broad outline. For example, Hurlbert & Jetz (2007, PNAS <https://doi.org/10.1073/pnas.0704469104>) found that Shapefile type range maps only accurately correspond to actual species occurrences at a resolution of about 200 km². Rasterizing range maps at 1 km² and assuming that species occur in each 1km² grid cell - and thus assuming that all those climates are within the species niche - risks overestimating the true extent of species NCNs. For example, when checking the range map of *Alces alces* (mention in Fig 1), it seems that the whole of Scandinavia is marked as part of the species native range. This includes all mountainous areas. Am I correct that in the current analyses, the climate conditions characterizing the 1 km² peak area of the Galdhøpiggen mountain (~2.500m, covered with eternal snow conditions) is considered to be within the species NCN? I agree with the authors that GBIF data can be strongly biased and likely only represent a part of the species actual niche, but I fear that this approach is biased in the opposite manner. A critical look at how native niches are defined here, and/or sensitivity analyses on the influence of the grain size at which native niches are quantified would help to strengthen my trust in the author's conclusions.

Reviewer #2:

Remarks to the Author:

Dear colleagues,

Your study on drivers of invasive mammals' establishment success is quite promising, straightforward and timely. In particular, the introduction of a new metric (niche margin index: NMI) may represent an important contribution to our toolset for studying biological invasions and, perhaps more importantly, predicting them. Therefore, I believe that your study can be certainly interesting for the broad readership of Nature Communications. That said, I do have some concerns on the presentation and contextualization of your study. Below, I provide comments that I hope can contribute to improve your study's potential.

First, I felt that your interesting findings were not properly discussed in light of the broader literature. Indeed, you properly cited several studies considering climatic matching but failed to discuss their findings in comparison with yours. For instance, considering the recent study on birds (your reference 16) that also considered climatic matching, are your findings similar/different? Can we draw the same conclusions across different taxa? I do believe that the broad readership of

Nature Communications deserves such comprehensive discussion.

Second, you are certainly aware that a species' native climatic niche represents, at most, its realized niche instead of its fundamental niche and this can have important implications on your findings and interpretations. For instance, the mere definition of niche margins can be severely affected by not considering "fundamental" niche margins but "realized" niche margins. Of course, measuring/describing or simply considering fundamental niches is not simple and I'm not suggesting that you do so here. However, I do believe that you should at least discuss this issue in your study (e.g. what is the main implication of not considering fundamental niche margins?).

Third, you claim that "it is...possible to predict establishment success using NCN-matching". However, you did not actually show such predictive capacity of your approach but instead show that you could explain establishment success very well. Would it be possible to test such "prediction capacity" with your data? Something akin to SDMs/ENMs evaluation (e.g. considering training and testing data or leaving data out for later validation)?

Finally, considering the authors' experience on code development, I expected them to provide reproducible code on how to derive the NMIs and conduct their analyses (sorry if I missed it, but couldn't see any reference to it). At least, code for NMI calculation would be highly desirable. Are the authors planning to include such code, for example, as part of existing R packages (e.g. ecospat)? Even if they plan to do so, it would be very useful for the reader to have such code in order to test/apply your novel approach. Indeed, providing such code (even if it is not accompany by the data used in this study) would allow a potential adoption of your approach for future studies. I strongly suggest that you do provide such code.

Minor issues (numbers correspond to your line numbering):

-line 50+. You would expect greater niche shifts as establishments fall further from the NCN margins, right? Could you test this prediction? I'm not suggesting that you do so here, just a potential idea for further analyses.

-line 65+. "niche truncation issues". This effect could also result from considering only the realized niche instead of the fundamental niche, correct? (See above). Perhaps mentioning this discrepancy between realized and fundamental niche would be quite informative for the reader.

-line 70. Is reference 19 the correct one here?

-line 140+. This assumes that those regions are in fact outside the species' fundamental niche, right?

-lines 155-160. Any idea on how would you go about testing this? Not that you should, but simply as a recommendation for future studies.

-line 160. "conditions not present in the native niche", but still within the species' fundamental niche?

-line 165+. "...depends on the particular combination of species attributes and release characteristics". Depends on or adds to the contribution of other factors?

-line 165+. Please, elaborate on how "[t]he results explain why general features of invasions have been difficult to characterize".

REVIEWER COMMENTS

Reviewer #1 (Remarks to the Author):

Species distribution models have become a common tool to predict sites at risk of invasion by introduced species. Yet, there is an ongoing debate about the reliability of these models, as studies frequently report mismatches between actual and predicted invasive occurrences. These mismatches are often interpreted as so-called 'niche shifts', calling into question the generality of the hypothesis of niche conservatism ('NCN', as termed by the authors) that underlies the use of species distribution models for forecasting invasive ranges.

The authors here point out that part of the debate may stem from the fact that the vast majority of published studies (necessarily) relies in invasive range occurrence data that represent a mix between areas that are occupied by invasive species because (a) species have been introduced there and they have successfully established a self-sustaining population, and (b) invasive spread from another location is the source of the invading individuals, not a new, independent release event. For case (b), it can be argued that invasive spread may results from e.g. adaptation to novel local climates, and that including such data when evaluation conservatism of

the native niche is not a 'fair' comparison. I think this is a very valuable remark, and commend the authors for gathering the data necessary to make a 'fair' assessment of the NCN.

*The authors then proceed to with a fair test of the NCN hypothesis using both a novel Niche Margin Index (NMI, see below) and more traditional Climate Suitability (CS) index derived from a species distribution model. Results were similar, evidencing that introductions are more likely to succeed when the area in which they were introduced had a climate that was more similar to the climate conditions species experience in their native range. This result corresponds with those of a recent meta-analysis of NCN studies (i.e. those studies that mix 'establishment' and 'invasive spread' occurrences'), which concludes that by and far, invasive species do conserve their climatic niche (Liu et al. 2020, PNAS, <https://doi.org/10.1073/pnas.2004289117>). For anticipating biological invasions, it is good news that both establishment success *sensu strictu* (and invasive spread seemingly too, Liu et al. 2020) are tightly linked to the NCN.*

> We thank the reviewer for this positive assessment of the utility of our work in the current discussion going on in the literature about the prediction of invasion risks using climate-matching to the NCN, and for suggesting this recent useful paper by Liu et al. 2020, which we have now added in the introduction.

One caveat may here however be that the authors use a rather liberal definition of a species native niche, reducing the chance that a non-native occurrence will fall outside of a species niche (see below, technical comments).

> This is a valid comment. We have responded to it in full details below, when addressing the related technical comment. We added a sensitivity study to assess this issue, without change to our results, and thus believe the revised manuscript now addresses it fully.

Assessing NCN has seen some methodologies put forward over the last decade, varying from metrics based on species distribution models or ordination frameworks, aimed at testing hypotheses on niche equivalency and similarity or on the proportion of species occurrences that are within or outside of a species native niche. While in theory these methods can be applied to 'establishment data' only, the authors here introduce an interesting novel metric, the NMI, that allows for each pixel in a geographical area to be assessed in terms of its 'distance' to the margins of the species native niche (negative outside, 0 at the margins, positive inside the niche).

While certainly an interesting metric, especially the 'niche outerness' component, the discussion of how this niche outerness can advance the field of biological invasions, both from a theoretical and applied perspective is a bit scant, in my opinion. The authors do mention that 'particular evolutionary or ecological processes [can] occur', but or as indication of climatic suitability under climate change, but little effort is made to relate their metric to concepts such as the 'tolerance niche', as defined by Sax et al. (2013, TREE 28: 517-523), or with potential relationships

between niche distances and population growth rates given in Holt et al Chapter 10 Theories of Niche Conservatism and Evolution: could exotic species be potential tests? pp 259-290 in Species Invasions: Insights into ecology, evolution and biogeography (eds. Sax, Stachowicz and Gaines), Sinauer Associates, Mass. This is a bit of a missed chance to showcase the relevance of this novel metric.

> We agree that the NMI metric could have been better related to the existing body of literature linking niche theory - and especially niche margins and the tolerance niche - and biological invasions. Thanks in this regard for pointing to these two valuable references by Holt and Sax, which we have now used to expand the related section in the discussion, adding studies by Pulliam and Godsoe, which were also particularly relevant.

Apart from these general remarks, I have only very few more technical comments, as the paper is overall well written, clearly structured and easy to follow.

> Thank you for your positive overall evaluation of the manuscript.

Abstract: maybe a bit nitpicking but I find the formulation that to be a successful invader, a species needs to establish in conditions within its native climate niche a 'bit upside down'. To be successful, all an invasive species needs to do is to establish a (self-sustaining) population. The hypothesis goes that such successful establishment will only be possible in areas that are within its native climatic niche.

> We agree that the order of the logical arguments was reversed. We have corrected the abstract accordingly.

Line 203-213: Were all occurrences (here: all pixels within native range) used to delineate the niche margins, as I see two potential problems here. (1) of the niche margins encompass even occurrences in very rare climate conditions, this may result in rather broadly, liberally defined 'native niches', making it more likely that invasive occurrence will fall within the native niche. Have you considered sensitivity analyses, e.g. calculating niche margins when omitting the, say, 5% occurrences that are in the most uncommon climates (climates here as 'pixels' in the 100-by-100 climate grid)? (2) This seems especially relevant for this study, as not Shapefile range maps have been used for characterizing the native distribution and thus the native niche. Range maps often do a pretty good job at describing the broad outlines of a species distribution, but this does not mean that species will occur at any given location within that broad outline. For example, Hurlbert & Jetz (2007, PNAS <https://doi.org/10.1073/pnas.0704469104>) found that Shapefile type range maps only accurately correspond to actual species occurrences at a resolution of about 200 km². Rasterizing range maps at 1 km² and assuming that species occur in each 1km² grid cell - and thus assuming that all those climates are within the species niche - risks overestimating the true extent of species NCNs. For example, when checking the range

map of *Alces alces* (mentioned in Fig 1), it seems that the whole of Scandinavia is marked as part of the species native range. This includes all mountainous areas. Am I correct that in the current analyses, the climate conditions characterizing the 1 km² peak area of the Galdhøpiggen mountain (~2.500m, covered with eternal snow conditions) is considered to be within the species NCN? I agree with the authors that GBIF data can be strongly biased and likely only represent a part of the species actual niche, but I fear that this approach is biased in the opposite manner. A critical look at how native niches are defined here, and/or sensitivity analyses on the influence of the grain size at which native niches are quantified would help to strengthen my trust in the author's conclusions.

> We agree that the way native niche margins are delineated is crucial and can potentially have a strong impact on the calculation of the NMI metric. Using 100% of occurrences to delineate the native niche margins can include climatic conditions that are rarely used by the species, making it more likely that invasive occurrences fall within the native niche. And we also agree that this potential issue may be exacerbated when occurrences are defined from range maps that may not be precise enough and include areas not occupied by the species (i.e. especially mountainous areas, indeed). In this regard, our approach effectively bears the risk of being too liberal. On the other hand, even rare habitats can still be used by the species, and thus indicate a real potential for establishment. In this regard, choosing a 100% envelope has the advantage of not depending on an arbitrary percentile threshold. We agree however that running sensitivity analyses is a proper way to assess this important issue. We accordingly redefined the niche margins using kernel density estimation functions from the *ks* package, which allowed us to obtain contour lines for specific levels of density. For all introductions, we recalculated NMI with level 99, 95 and 90% and reran the Bayesian mixed models for each level (SOM Figs. S7 to S10). The importance of NMI in the mixed model was indeed even more strongly supported with the use of the new kernel (effect size increased from 0.22 to 0.31). We further conducted a sensitivity analysis using climatic datasets aggregated at resolutions of 10', 0.5° and 1°. In the main text we present the results with a level of density of 99% and a resolution of 10', and all sensitivity analyses can be found in the supplementary material (SOM Figs. S7 to S10). We show that the resolution of the dataset does not influence the results significantly, but there was a trend for more negative NMI values with smaller percentage levels (and marginally smaller for the dataset at 1° resolution). Importantly, in all cases, the association between NMI and introduction success remains significant with all sets of parameters, and therefore the new analyses did not change our initial conclusions.

Reviewer #2 (Remarks to the Author):

Dear colleagues,

Your study on drivers of invasive mammals' establishment success is quite promising, straightforward and timely. In particular, the introduction of a new metric (niche margin index:

NMI) may represent an important contribution to our toolset for studying biological invasions and, perhaps more importantly, predicting them. Therefore, I believe that your study can be certainly interesting for the broad readership of Nature Communications. That said, I do have some concerns on the presentation and contextualization of your study. Below, I provide comments that I hope can contribute to improve your study's potential.

> Thank you for the enthusiastic comment on the usefulness of the approach and the new metric we developed. We respond below to your concerns on the presentation and contextualization of the study.

First, I felt that your interesting findings were not properly discussed in light of the broader literature. Indeed, you properly cited several studies considering climatic matching but failed to discuss their findings in comparison with yours. For instance, considering the recent study on birds (your reference 16) that also considered climatic matching, are your findings similar/different? Can we draw the same conclusions across different taxa? I do believe that the broad readership of Nature Communications deserves such comprehensive discussion.

> Thanks for the useful comment. We agree that it deserved more attention and have expanded the discussion accordingly.

Second, you are certainly aware that a species' native climatic niche represents, at most, its realized niche instead of its fundamental niche and this can have important implications on your findings and interpretations. For instance, the mere definition of niche margins can be severely affected by not considering "fundamental" niche margins but "realized" niche margins. Of course, measuring/describing or simply considering fundamental niches is not simple and I'm not suggesting that you do so here. However, I do believe that you should at least discuss this issue in your study (e.g. what is the main implication of not considering fundamental niche margins?).

> Thanks for this very valid comment and suggestion. We are indeed very aware that the niche we consider here is the realized one, and thus it is true that we accordingly assess the realized niche margins, which could change for a species when introduced in an exotic range (compared to the native range), where e.g. competitors are missing. We agree this wasn't sufficiently clear in the previous version of the manuscript, and we have now clarified in the introduction that we implicitly quantify the realized niche. We also fully agree that it would be interesting to run such NCN matching with the fundamental niche, but as you mention, measuring it is far from an easy task and not possible for most species (i.e. it must be done under controlled experiments), which also explains why most niche studies are based on the realized niche. Yet, we agree that this issue deserves more discussion, which we have now developed according to your suggestion. Furthermore, we now provide results for niche margins delineated with a minimum volume ellipsoid (MVE). This ellipse is centered on the maximum density of

occurrence, and small and large axes are calibrated to encompass a defined percentage of occurrences (i.e. we provide results for 99, 95 and 90%). This approach has been proposed as a proxy to the fundamental niche (Escobar et al. 2018, *Ecology & Evolution*). Results based on the MVE show that NMI is shifted towards smaller values, but the main pattern remains.

Third, you claim that “it is...possible to predict establishment success using NCN-matching”. However, you did not actually show such predictive capacity of your approach but instead show that you could explain establishment success very well. Would it be possible to test such “prediction capacity” with your data? Something akin to SDMs/ENMs evaluation (e.g. considering training and testing data or leaving data out for later validation)?

> Our primary aim was indeed to explain establishment success as a function of NMI and other life-history factors. For this reason, we initially did not test the predictive capacity of the approach using an independent set of introductions. We have now run another set of models calibrated with 80% of the occurrences and tested their predictive capacity on the 20% remaining occurrences using a repeated split-sampling approach. We obtained a fairly high value of predictive power (maxTSS = 0.52), yet maybe limited by differences in response between species. Fitting species-specific models with relevant explanatory variables could therefore likely yield greater predictive power. We now address these predictive capacity issues in the discussion.

Finally, considering the authors’ experience on code development, I expected them to provide reproducible code on how to derive the NMIs and conduct their analyses (sorry if I missed it, but couldn’t see any reference to it). At least, code for NMI calculation would be highly desirable. Are the authors planning to include such code, for example, as part of existing R packages (e.g. ecospat)? Even if they plan to do so, it would be very useful for the reader to have such code in order to test/apply your novel approach. Indeed, providing such code (even if it is not accompany by the data used in this study) would allow a potential adoption of your approach for future studies. I strongly suggest that you do provide such code.

> We agree that providing the code is very important for scientific reproducibility. The reference to the github repository was indeed given (<https://github.com/ecospat/NMI>), in which you will find the NMI function itself, the code to run the NMI analyses and the code to create Figs 1c, 2b, 3 and S5. We apologize if it was not sufficiently visible and have now added a specific “code availability” section in the main text.

Minor issues (numbers correspond to your line numbering):

-line 50+. You would expect greater niche shifts as establishments fall further from the NCN margins, right? Could you test this prediction? I’m not suggesting that you do so here, just a potential idea for further analyses.

> Yes, a population establishing successfully far from the NCN margin is likely to contribute to a big niche shift. Yet we do not have the proper data to develop on this aspect here, since we do not have the full distribution in the invaded range. However, our previous work on the spotted knapweed (Broennimann et al. 2014 *J. Biogeogr.*), which shows two contrasting invasion patterns in North America, indicates that niche shifts can happen even when the first establishment site is within the NCN if it is surrounded by conditions different from the NCN (shown in the spotted knapweed example by the invasion of the West coast). Thanks for the suggestion. We now mention this in the discussion, also proposing that future studies could attempt to link our results with findings on niche shifts for the same species.

-line 65+. "niche truncation issues". This effect could also result from considering only the realized niche instead of the fundamental niche, correct? (See above). Perhaps mentioning this discrepancy between realized and fundamental niche would be quite informative for the reader.

> Yes, although we here refer to another type of 'truncation', we agree it could also happen due to the realized niche being considered. We now mention in the introduction that we consider the realized niche and we better discuss the fundamental versus realized niche aspects in the introduction and discussion (also in the light of the additional niche references cited; see our previous response in this regard).

-line 70. Is reference 19 the correct one here?

> It should have been reference 10 (Bomford et al. 2009). Thank you for pointing to this mistake.

-line 140+. This assumes that those regions are in fact outside the species' fundamental niche, right?

-lines 155-160. Any idea on how would you go about testing this? Not that you should, but simply as a recommendation for future studies.

-line 160. "conditions not present in the native niche", but still within the species' fundamental niche

> These regions with negative NMI might be outside the species' fundamental niche, but as we delineated the NCN from occurrence data, we can only guarantee that they are outside of the realized native niche. As mentioned above, delineating the fundamental climatic niche is hardly possible for most species. We have however considered minimum volume ellipsoids (MVE) to delineate a best proxy of the fundamental niche (see main response on fundamental vs. realized niche above). This analysis shows that 5% of successful establishments with a negative NMI based on the realized NCN (kde

approach) remain negative when based on MVE (Fig. S11), indicating that these populations were potentially introduced outside their fundamental niche (and thus in the tolerance niche defined by Sax et al. 2013 *TREE*). Yet, the MVE, by being still based on realized native populations, remains a very rough estimate likely underestimating the fundamental niche. We now discuss this aspect in the manuscript, along with the idea of the tolerance niche.

-line 165+. "...depends on the particular combination of species attributes and release characteristics". Depends on or adds to the contribution of other factors?

> NMI adds to the contribution of other factors. We rephrased the sentence to improve clarity.

-line 165+. Please, elaborate on how "[t]he results explain why general features of invasions have been difficult to characterize".

> We now cite Pyšek et al. 2020 *Neobiota*, who propose a framework to disentangle large-scale context dependence in biological invasions.

Reviewers' Comments:

Reviewer #1:

Remarks to the Author:

Dear authors,

Many thanks for this revised ms, where all my comments have been adequately addressed. The sensitivity analyses conducted indeed confirm the validity of the proposed approach, and the discussion now better links to findings to the relevant literature. I thus have no further comments on the ms.

Reviewer #2:

Remarks to the Author:

Dear colleagues,

I appreciate the effort to accommodate all of my previous criticisms and suggestions. I was particularly satisfied with the inclusion of sensitivity analyses, the consideration of ellipsoids to approximate fundamental niches of species, and the provision of a broader discussion. More importantly, I believe that such improvements can certainly strengthen the potential relevance of your study.

Kind regards,

Fabricio Villalobos